# Shear Bearing Capacity of Steel-Fiber-Reinforced Concrete Shear Wall under Low-Cycle Repeated Loading Based on the Softened Strut-and-Tie Model

**Peibo You [1], Jie Zhang [2], Binyu Wang [3], Yi Wang [1], Qingjie Yang [1] and Li Li [3],***

[1] Department of Civil and Transportation Engineering, Henan University of Urban Construction, Pingdingshan 467036, China; 30070513@huuc.edu.cn (P.Y.); 30010615@huuc.edu.cn (Y.W.); 15093878183@163.com (Q.Y.)

[2] Department of Architectural Engineering, Jiyuan Vocational and Technical College, Jiyuan 459000, China; hylzj6@126.com

[3] College of Water Resources and Architectural Engineering, Northwest A&F University, Yangling 712100, China; nwafu-wang@nwafu.edu.cn

\* Correspondence: drlili@nwafu.edu.cn

**Abstract:** In this paper, the loading mechanism of steel-fiber-reinforced concrete (SFRC) shear wall (SW) under low-cycle repeated loading is analyzed, and the softened strut-and-tie model (SSTM) of SFRC SW composed of horizontal and vertical resistant members and diagonal strut is proposed, in which the contributions of distributed web reinforcement, concrete, and steel fiber (SF) to the shear bearing capacity (SBC) of SFRC SW is identified. Furthermore, a new algorithm to obtain the SBC of SFRC SW is established, and then it is validated by using the test results of steel-fiber-reinforced high-strength concrete (SFHSC) SW and SFRC SW under low-cycle repeated loading. The results show that the calculated values are in good agreement with the experimental values for the 11 SFRC SWs, and the average strength ratio between calculated and experimental values ($V_{jh,t}/V_{jh,c}$) is 0.958. Therefore, the proposed calculation method is scientific and accurate for analyzing and predicting the SBC of SFRC SW. In addition, the proposed calculation method can scientifically and accurately analyze and predict the SBC of SFRC SW.

**Keywords:** steel fiber; shear wall; softened strut-and-tie model; shear bearing capacity



## 1. Introduction

Reinforced concrete (RC) shear wall (SW) is a common anti-lateral force component in high-rise buildings that has been widely used in high-frequent earthquake-prone areas [1–5]. The observation from recent earthquake reconnaissance indicated that the main reason why the RC SW was seriously damaged and even collapsed was its inadequate ductility and energy dissipation [6–8]. There are many problems that need to be solved urgently in the existing ordinary reinforced concrete SW located in high-intensity earthquake areas. In the existing design code, it is imperative to tightly control the axial compression ratio of RC SW at the base of high-rise buildings. This strict limitation is essential for adhering to ductility requirements and preventing brittle failure. Consequently, the wall web is frequently designed to be excessively thick. This not only diminishes the available area and space within the building but also leads to increased structural self-weight and construction expenses. On the other hand, ordinary RC SWs necessitate a substantial number of stirrups in the restrained edge members. These stirrups are essential to effectively confine the concrete and prevent the longitudinal reinforcement from buckling when subjected to compression. However, intensive reinforcement of RC SWs not only increases construction cost, but also affects construction quality.

The research shows that there are two main reasons for the insufficient ductility and energy dissipation performance of ordinary RC SWs. First, the longitudinal reinforcements

in the restrained edge members of RC SWs yield prematurely, which leads to rapid degradation of the flexural capacity of RC SWs. Second, the SBC of the wall degrades rapidly after the concrete at the bottom of the RC SW is crushed and peeled off, which reduces the energy dissipation capacity of the RC SW [9,10]. In order to improve the ductility and energy dissipation performance of ordinary RC SWs, some new improvement measures have been put forward one after another, such as improving the reinforcement ratio and increasing the numbers of wall reinforcements, SWs with vertical joins, SWs with horizontal joints, SWs with concealed bracing, steel-reinforced concrete SWs, and reinforcements with different fibers, or even changing the composition of concrete [11–19]. However, the improvement of ductility and energy dissipation capacity of RC SWs cannot simply rely on increasing the distribution reinforcement ratio, which will increase the construction difficulty. Additionally, the joints in the wall web easily weaken the SW section, and the addition of concealed bracing in the wall web easily leads to complexity of the wall reinforcement mode. Furthermore, the wall cracks cannot be effectively controlled by profiled steel set up in both ends of the RC wall as embedded columns or a steel plate brace precasted in the SW.

In particular, as high-strength concrete (HSC) has become more commonly used, it allows for a reduction in the cross-sectional size of the SW. However, this reduction in size comes with an inevitable increase in the brittleness of reinforced high-strength concrete (RHSC) SWs. Correspondingly, the seismic behavior of RHSC SWs needs to be improved to meet the higher requirements [20,21]. The latest research results of scholars at home and abroad showed that adding SF into concrete can effectively improve the seismic performance of RC members, including the RHSC SW [22–24]. The seismic behavior of SFRC SW subjected to reversed cyclic loading had been tested by several scholars [3,25–27]. Just as anticipated, because the randomly distributed SF can effectively improve anti-cracking properties, tensile strength, shearing property, toughness, durability, and seismic performance of matrix concrete, then they can greatly improve the seismic performance of RC SWs, reduce the degree of reinforcement aggregation, and increase construction efficiency [28]. With the increase in the volume ratio of SF, the bearing capacity, ductility, and energy dissipation capacity of SFRC SWs increase. Hitherto, extensive research results have been obtained from experimental studies on the seismic performance of RC SWs and SFRC SWs, but the research results on the mechanical mechanism and calculating method for SBC of SFRC SWs were relatively few since the mechanical mechanism of SWs subjected to the combined action of bending, compression, and shear is very complicated. What is more, how to consider the role of SF in concrete is also a complicated problem. Additionally, the calculating methods for SBC of SFRC SWs proposed in the existing literature are almost universally half-empirical formulas based on test results, and there is no scientific and reasonable theoretical model. Furthermore, ultimately, the calculation results for SBC of SFRC SWs were not accurate enough. Therefore, there is an urgent need to propose scientifically sound theoretical models. Some scholars have proposed models and theories for calculating the SBC of SWs, such as Panatchai et al. [29], who predicted the peak shear strength of squatting SWs with and without boundary elements by developing a strut-and-tie model (STM); and Oudah et al. [30], who proposed the theory of two-discrete-elements (TDE) shear deformation for evaluating the deformability of RC SWs under lateral loads in earthquake-resistant design applications.

The SSTM for calculating the SBC of RC elements damaged by inclined compression bar was established by S.J. Hwang and H.J. Lee, which derived from the concept of the strut and tie of RC elements, introducing a softening coefficient of concrete. Furthermore, the equations satisfying the equilibrium, constitutive relation, and coordination of cracked RC were put forward. The SBC prediction of different types of RC members has been calculated by using the SSTM, and the accuracy of these calculations has been confirmed through a comparison of the calculated SBC with test results reported in the previous literature [31–34].

In this paper, the loading mechanism of SFRC SWs was analyzed; the randomly distributed SF in the SW web could be equivalent to horizontal and vertical finely distributed steel bars. Then the SSTM to calculate the SFRC SW composed of diagonal strut, horizontal, and vertical resistant members was proposed, in which the contributions of SF, concrete, and distributed web reinforcement to the SBC of SFRC SW was identified. Furthermore, a new algorithm based on the SSTM to obtain the SBC of SFRC SWs was established, and then it was validated by using the test results of 11 low-rise SFHSC and SFRC SWs under low-cycle repeated loading. The results indicated a favorable agreement between the calculated values and experimental data for the 11 low-rise SWs, and the SSTM may be used to calculate the SBC of SFHSC and SFRC SWs. Based on the analysis of the loading mechanism of SFRC SWs, a well-established calculation method for the SBC of SFRC SWs was proposed. It has a better theoretical basis and a more scientific and reasonable theoretical model than the previous calculation method. Finally, the calculation results for the SBC of SFRC SWs will be accurate enough.

## 2. Loading Mechanism of SFRC SW

While the loading mechanism of SFRC SWs under the combined forces of bending, compression, and shear is highly intricate, the random distribution of SF within the SW web can be deemed an equivalent alternative to the placement of finely distributed horizontal and vertical steel bars. Notably, SF do not alter the SW's loading mechanism, thereby making it akin to that of RC SWs. The wall forms intersecting oblique pressure and tension flow along the diagonal direction, as shown in Figure 1. The diagonal compressive struts are formed by the wall web matrix concrete under pressure flow. The diagonal tension flow in the wall web is mainly supported by SFRC due to its good tensile and crack resistance prior to the wall web cracking, and during this stage, the reinforcement stress can be negligible. As the load increases, SFRC will develop initial oblique cracks when the diagonal tension flow in the wall web exceeds the crack strength of SFRC. Once SFRC displays the first crack, the diagonal tension flow in the wall web is carried by the strut and tie. The horizontal tie rod contains SF and horizontally distributed reinforcements, as depicted in Figure 1b, while the vertical tie comprises SF and vertically distributed reinforcements, as shown in Figure 1c. To summarize, when the initial crack occurred in the wall web, the randomly dispersed steel fibers and the distributed reinforcements primarily withstood the tensile forces, while the concrete experienced oblique compressive stresses. This combination of forces resulted in the development of a strut-and-tie action. Therefore, the SSTM of SFRC SWs also involved diagonal struts, as well as horizontal and vertical resisting components.

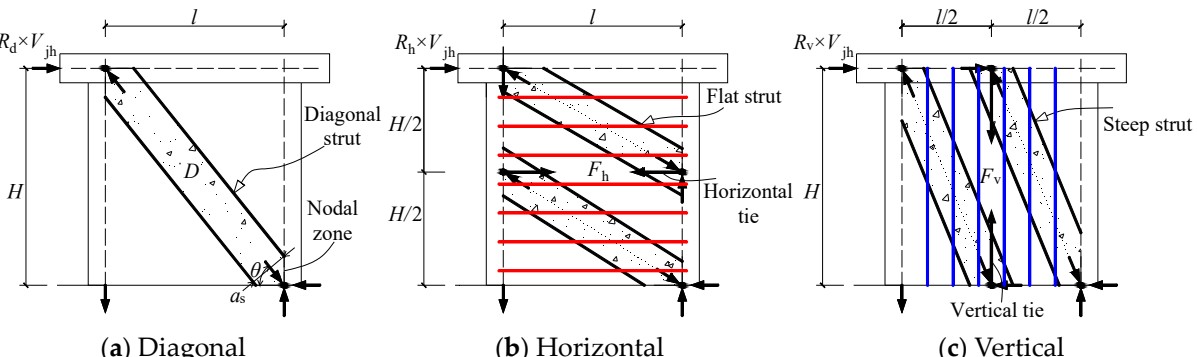

| (a) Diagonal | (b) Horizontal | (c) Vertical |

**Figure 1.** Shear resisting mechanisms of shear wall.

In Figure 1a, the diagonal force mechanism in SFRC is established by a single diagonal compression strut. The inclination angle $\theta$ of this SFRC diagonal compression strut can be determined by [35]:

$$\theta = \frac{1}{\tan\left(\frac{H}{l}\right)} \tag{1}$$

where $H$ is the height of the SFRC SW; $l$ is internal force arm of vertical couple of the SFRC SW and $l$ is equal to $0.9h$; and $h$ is the section height of the SFRC SW.

The cross-sectional area of the diagonal strut, $A_{\text{str}}$, can be determined as [36]:

$$A_{\text{str}} = \alpha_{\text{str}} \times b_{\text{str}} \tag{2}$$

where $\alpha_{\text{str}}$ is the section height of the diagonal strut; and $b_{\text{str}}$ is the section width of the diagonal strut, which is consistent with the wall section width $b$.

The section height of diagonal strut $\alpha_{\text{str}}$ can be roughly defined as [37]:

$$\alpha_{\text{str}} = (0.25 + 0.85\frac{N}{bhf_{\text{c}}'})h \tag{3}$$

where $N$ is the axial force borne by the SFRC SW specimen; and $f_{\text{c}}'$ is the axial compressive strength of the concrete cylinder (MPa).

Figure 1b depicts a horizontal force mechanism incorporating a single tie rod and two flat struts. The presence of SF improved the tensile, crack, and toughness properties of concrete. Accordingly, SFRC SWs performed better than RC SWs. Thus, the horizontal tie rod of the SFRC SW web comprised randomly distributed SF and horizontally distributed reinforcements, and can be defined as [37]:

$$F_h = F_{h,sf} + F_{h,s} \tag{4}$$

where $F_h$ is the horizontal tie rod value, $F_{h,s}$ is the tension force value of the horizontally distributed reinforcement tie rod. Numerous test results indicated that the horizontally distributed reinforcements in the wall web did not undergo full yielding when the SW specimen experienced failure. Therefore, $\eta_1$ is suggested as the effective coefficient in the background. Thus, $F_{h,s} = \eta_1 \times A_{h,s}$, $\eta_1$ is the effective shear resistance coefficient of horizontally distributed reinforcement and can be approximated as 0.75, as reported in the literature [34]; $A_{h,s}$ is the cross-sectional area of the horizontally distributed reinforcement tie rod; $f_{h,s}$ is the tension strength of horizontally distributed reinforcement; $F_{h,sf}$ is the tension force value of the horizontally distributed SF tie rod; in $F_{h,sf} = A_{h,sf} \times f_{sf}$, $A_{h,sf}$ is the cross-sectional area of the horizontally distributed SF tie rod; and $f_{\text{sf}}$ is the tension strength of horizontally distributed SF.

In order to reduce the cost of computation, the random distribution of SF within the three dimensions of the SFRC SW web can be considered equivalent to the arrangement of horizontally and vertically finely distributed steel bars in the SBC analysis of SFRC SWs. Consequently, the cross-sectional area of the horizontally distributed SF tie rod, $A_{h,sf}$, can be determined by [38]:

$$A_{h,sf} = n_{sf} A_{sf} \tag{5}$$

where $A_{sf}$ is the cross-sectional area of a single SF; and $n_{sf}$ is the number of equivalent horizontal SFs, which can be determined by [38]:

$$n_{sf} = \eta_2 \rho_f \frac{bH}{A_{sf} \sin \theta} \tag{6}$$

where $\rho_f$ is the volume ratio of SF; the equivalent reduction coefficient, denoted as $\eta_2$, can be approximately considered as 0.41, as inferred from the analysis of experimental results reported in the literature [38].

The cross-sectional area of the horizontally distributed SF tie rod, $A_{h,sf}$, can be calculated by:

$$A_{h,sf} = 0.41\rho_f bH / \sin \theta \tag{7}$$

In Figure 1c, a vertical tie rod and two steep struts form the vertical force mechanism. Moreover, like the horizontal tie rod, the vertical tie rod of the SFRC SW web includes

randomly distributed SFs and vertically distributed reinforcements, and can be defined as the following [37]:

$$F_v = F_{v,sf} + F_{v,s} \qquad (8)$$

where $F_v$ is the tension force value of the vertical tie rod value; $F_{v,s}$ is the tension force value of the vertically distributed reinforcement tie rod. Numerous test results indicated that the vertically distributed reinforcements in the wall web did not undergo full yielding when the SW specimen experienced failure, so $\eta_3$ is suggested as the effective coefficient in the background, thus in $F_{v,s} = \eta_3 \times A_{v,s} \times f_{v,s}$, $\eta_3$ is the effective shear resistance coefficient of vertically distributed reinforcement and it can be approximated as 0.80, as reported in the literature [34]; $A_{v,s}$ is the cross-sectional area of the vertically distributed reinforcement tie rod, $f_{v,s}$ is the tension strength of vertically distributed reinforcement; and $F_{v,sf}$ is the tension force value of the vertically distributed SF tie rod. In $F_{v,sf} = A_{v,sf} \times f_{sf}$, $A_{v,sf}$ is the cross-sectional area of the vertically distributed SF tie rod and for $A_{v,sf} = 0.41\rho_f bh/cos\theta$, $f_{sf}$ is the tension strength of vertically distributed SF.

## 3. Computation Method for SBC of SFRC SW

### 3.1. Equilibrium Equations

Figure 2 shows the calculation diagram of the strut-and-tie model for the SFRC SW specimen. The horizontal and vertical shear forces of the SFRC SW specimen on the basis of the above strut-and-tie model can be defined as the following [37]:

$$\begin{aligned} V_{jh} &= D\cos\theta + F_h + F_v \cot\theta \\ V_{jv} &= D\sin\theta + F_v + F_h \tan\theta \end{aligned} \qquad (9)$$

where $D$ is the pressure value of the SFRC diagonal compression strut; $V_{jh}$ is the horizontal shear resistance capacity value of the SFRC SW web; and $V_{jv}$ is the vertical shear resistance capacity value of the SFRC SW web.

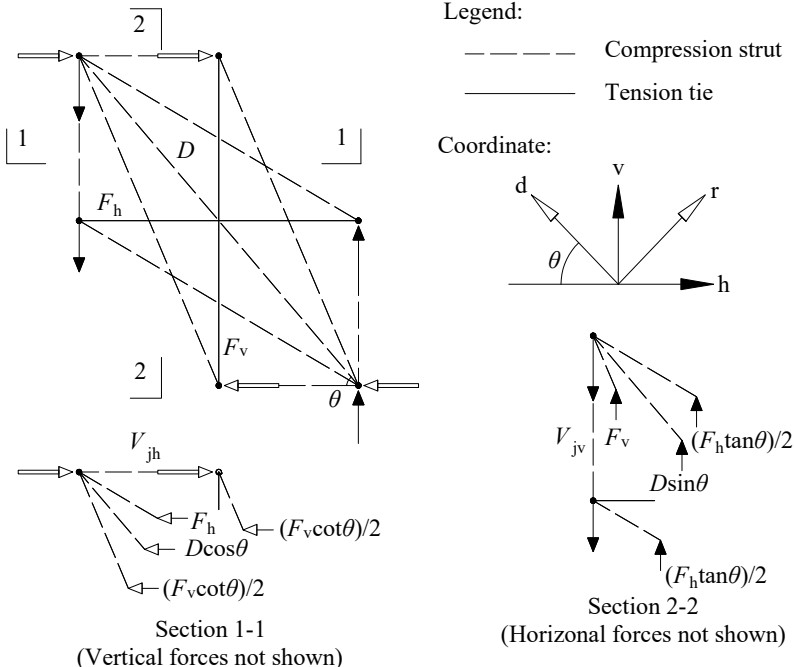

**Figure 2.** Strut-and-tie model of the SFRC shear wall specimen.

The horizontal shear resistance capacity value of the SFRC SW web, $V_{jh}$, is distributed to the three resistance mechanisms in a certain proportion [37]:

$$D \cos \theta : F_h : F_v \cot \theta = R_d : R_h : R_v \tag{10}$$

where $R_d$, $R_h$, and $R_v$ are the horizontal shear resistance capacity ratios borne by the diagonal, horizontal, and vertical resistance mechanisms, respectively, which can be calculated by [37]:

$$
\begin{aligned}
R_d &= \frac{(1-\gamma_h)(1-\gamma_v)}{1-\gamma_h\gamma_v} \\
R_h &= \frac{\gamma_h(1-\gamma_v)}{1-\gamma_h\gamma_v} \\
R_v &= \frac{\gamma_v(1-\gamma_h)}{1-\gamma_h\gamma_v}
\end{aligned}
\tag{11}
$$

where $\gamma_h$ is the ratio of the horizontal shear resistance capacity value borne by the horizontal tie rod when the vertical resistance mechanism does not participate in shear resistance capacity distribution; $\gamma_v$ is the ratio of the vertical shear resistance capacity value borne by the vertical tie rod when the horizontal resistance mechanism does not participate in shear resistance capacity distribution. The values of $\gamma_h$ and $\gamma_v$ can be calculated by [39]:

$$
\begin{aligned}
\gamma_h &= \frac{2\tan\theta-1}{3}, \ 0 \leq \gamma_h \leq 1 \\
\gamma_v &= \frac{2\cot\theta-1}{3}, \ 0 \leq \gamma_v \leq 1
\end{aligned}
\tag{12}
$$

The values of $D$, $F_h$, and $F_v$ can be calculated through Equations (9) and (10) as:

$$
\begin{aligned}
D &= \frac{1}{\cos\theta} \times \left(\frac{R_d}{R_d+R_h+R_v}\right) \times V_{jh} \\
F_h &= \left(\frac{R_h}{R_d+R_h+R_v}\right) \times V_{jh} \\
F_v &= \tan\theta \times \left(\frac{R_v}{R_d+R_h+R_v}\right) \times V_{jh}
\end{aligned}
\tag{13}
$$

The failure criterion of SSTM for the SFRC SW specimen is that the resultant force of the diagonal compression strut, flat compression strut, and steep compression strut at the joint region reaches the concrete compression strength, as shown in Figure 1. In order to judge whether the SFRC SW specimen is damaged, the resultant force at the joint region must be checked as shown in Figure 2. The maximum compressive stress $\sigma_{d,\max}$ generated by the three compression struts at the joint region can be defined as [37]:

$$\sigma_{d,\max} = \frac{1}{A_{\text{str}}}\left[D + \frac{F_h}{\cos\theta}\left(1 - \frac{\sin^2\theta}{2}\right) + \frac{F_v}{\sin\theta}\left(1 - \frac{\cos^2\theta}{2}\right)\right] \tag{14}$$

*3.2. Constitutive Equations*

The compressive softened stress–strain relationship of cracked SFRC can be expressed as [40]:

$$\sigma_d = \zeta f_c'\left[2\left(\frac{\varepsilon_d}{\zeta\varepsilon_0}\right) - \left(\frac{\varepsilon_d}{\zeta\varepsilon_0}\right)^2\right], \ \frac{\varepsilon_d}{\zeta\varepsilon_0} \leq 1 \tag{15}$$

where $\sigma_d$ is the average principal compressive stress of SFRC in the d-direction; $\zeta$ is the softened coefficient of SFRC; $\varepsilon_d$ is the average principal compressive strain corresponding to stress $\sigma_d$; and $\varepsilon_0$ is the strain of the SFRC cylinder when the stress of that reaches $f_c'$, which can be calculated approximately by [41]:

$$\varepsilon_0 = -0.002 - 0.001\left(\frac{f_c' - 20}{80}\right), \ 20 \ \text{MPa} \leq f_c' \leq 100 \ \text{MPa} \tag{16}$$

The softened coefficient of SFRC, $\zeta$, can be determined by the following [40]:

$$\zeta = \frac{5.8}{\sqrt{f_c'}}\frac{1}{\sqrt{1 + 400\varepsilon_r}} \leq \frac{0.9}{\sqrt{1 + 400\varepsilon_r}} \tag{17}$$

where $\varepsilon_r$ is the average principal tensile strain corresponding to the average principal tensile stress ($\sigma_r$) of the SFRC in the r-direction.

The stress $\sigma_d$ and the strain $\varepsilon_d$ need to meet the following conditions when the SFRC SW reaches the maximum SBC [40]:

$$\sigma_d = \zeta f_c' \tag{18}$$

$$\varepsilon_d = \zeta\varepsilon_0 \tag{19}$$

The stress–strain relationship of the distributed reinforcement in the SFRC SW can be expressed as [36]:

$$\begin{cases} f_s = E_s\varepsilon_s & \varepsilon_s < \varepsilon_y \\ f_s = f_y & \varepsilon_s \geq \varepsilon_y \end{cases} \tag{20}$$

where $E_s$ is the elastic modulus of the distributed reinforcements; $f_y$ and $\varepsilon_y$ are the yield strength and yield strain of the distributed reinforcements, respectively; and $f_s$ and $\varepsilon_s$ are the actual stress and strain of the distributed reinforcements, respectively; the only caveat here is that when the calculation Equation (20) above is applied to the horizontally distributed reinforcements and the vertically distributed reinforcements respectively, $f_s$ is taken as $f_{h,s}$ or $f_{v,s}$, $E_s$ is taken as $E_{h,s}$ or $E_{v,s}$, $\varepsilon_s$ is taken as $\varepsilon_{h,s}$ or $\varepsilon_{v,s}$, and $f_y$ is taken as $f_{yh}$ or $f_{yv}$.

The stress–strain relationship of the randomly distributed SF in the three dimensions of the SFRC SW web can be described as [36]:

$$f_{sf} = E_{sf}\varepsilon_{sf} \tag{21}$$

where $E_{sf}$ and $\varepsilon_{sf}$ represent the elastic modulus and actual strain of the randomly distributed SF in the three dimensions of the SFRC SW web, respectively.

Existing test results have indicated that, in the wall web, most of the randomly distributed steel fibers are pulled out from the matrix concrete rather than being damaged due to their good mechanical properties. Therefore, the tension strength of the randomly distributed SF in the wall web, $f_{sf}$, is primarily influenced by the bonding strength between the SF and matrix concrete, and the following functional relationship needs to be satisfied as [42]:

$$A_{sf}f_{sf} \leq \lambda_{sf}A_{spf}\tau_{sf,\max} \tag{22}$$

where $\lambda_{sf}$ is the effective coefficient of SF type, and the values of $\lambda_{sf}$ for the type of long straight, wave-shaped, and hooked SF are 0.5, 0.75, and 1.0, respectively; $\tau_{sf,max}$ is the maximum bonding strength between SF and matrix concrete according to the existing research results in the literature [42]; $\tau_{sf,max}$ can be considered equal to 2.5 times $f_{ct}$, where $f_{ct}$ represents the matrix tension strength of SFRC; $A_{spf}$ represents the surface area of SF; in $A_{spf} = \pi d_f l_{sfo}$, $d_f$ and $l_{sfo}$ are the equivalent diameter and the effective bounding length of SF respectively; and in $l_{sfo} = 0.25 \ l_f$, $l_f$ is the length of SFs.

Thus, the tension strength of the randomly distributed SF in the wall web, $f_{sf}$, can be described as the following according to Equation (22):

$$f_{sf} \leq \lambda_{sf} \left( \frac{l_f}{d_f} \right) \tau_{sf,\max} \tag{23}$$

In summary, the relationship between tension force values and strains of the tie rod can be expressed as:

$$F_h = F_{h,s} + F_{h,sf} = 0.75 A_{h,s} E_{h,s} \varepsilon_{h,s} + 0.41 \rho_f b H E_{sf} \varepsilon_{h,sf} / \sin\theta \leq F_{yh} \tag{24}$$

$$F_v = F_{v,s} + F_{v,sf} = 0.8 A_{v,s} E_{v,s} \varepsilon_{v,s} + 0.41 \rho_f b h E_{sf} \varepsilon_{v,sf} / \cos\theta \leq F_{yv} \tag{25}$$

$$\begin{aligned} \varepsilon_{h,s} &= \varepsilon_{h,sf} = \varepsilon_h \\ \varepsilon_{v,s} &= \varepsilon_{v,sf} = \varepsilon_v \end{aligned} \tag{26}$$

where $\varepsilon_{h,s}$ and $\varepsilon_{h,sf}$ are the strains of the horizontally distributed reinforcements and SF, respectively; $\varepsilon_{v,s}$ and $\varepsilon_{v,sf}$ are the strains of the vertically distributed reinforcements and SF, respectively; $\varepsilon_h$ and $\varepsilon_v$ are the average horizontal strain and the average vertical strain of the SFRC wall web, respectively; and $F_{yh}$ and $F_{yv}$ are the yield force values of the horizontal and vertical tie rods, respectively.

### 3.3. Compatibility Equations

The strain compatibility equations used in this research are shown as follows [43]:

$$\varepsilon_r = \varepsilon_h + (\varepsilon_h - \varepsilon_d)\cot^2\theta \tag{27}$$

$$\varepsilon_r = \varepsilon_v + (\varepsilon_v - \varepsilon_d)\tan^2\theta \tag{28}$$

### 3.4. Solution Steps

The proposed solution steps for the SBC of SFRC SWs are shown in Figures 3 and 4, and the main solving steps are as follows: at first, the horizontal shear resistance capacity value of the SFRC SW, $V_{jh}$, is assumed, and the values of $D$, $F_h$, $F_v$, and $\sigma_{d,max}$ can be derived by solving the Equations (9)–(14). Subsequently, the softened coefficient of the SFRC, $\xi$, is initially determined using Equation (18), assuming that the concrete diagonal strut reaches its compressive strength. Next, the strains of the tension and compression struts are obtained by using the corresponding constitutive equations. At last, the average principal tensile strain in the r-direction, $\varepsilon_r$, is determined using compatibility equations. Then, a new value of $\xi$ is obtained through the Equation (17). If the initial value of the softened coefficient of SFRC, $\xi$, closely approximates the new value, $\xi$, then the assumed value, $V_{jh}$, represents the SBC of the SFRC SW. Otherwise, back to the iterations.

However, it is important to realize that the solution flow in Figure 3 is categorized into five calculation types based on different yield conditions of the tie rods [34], as illustrated in Table 1. Figure 4 depicts the stress redistribution of shears of the SFRC SW web after the horizontal tie rod has yielded.

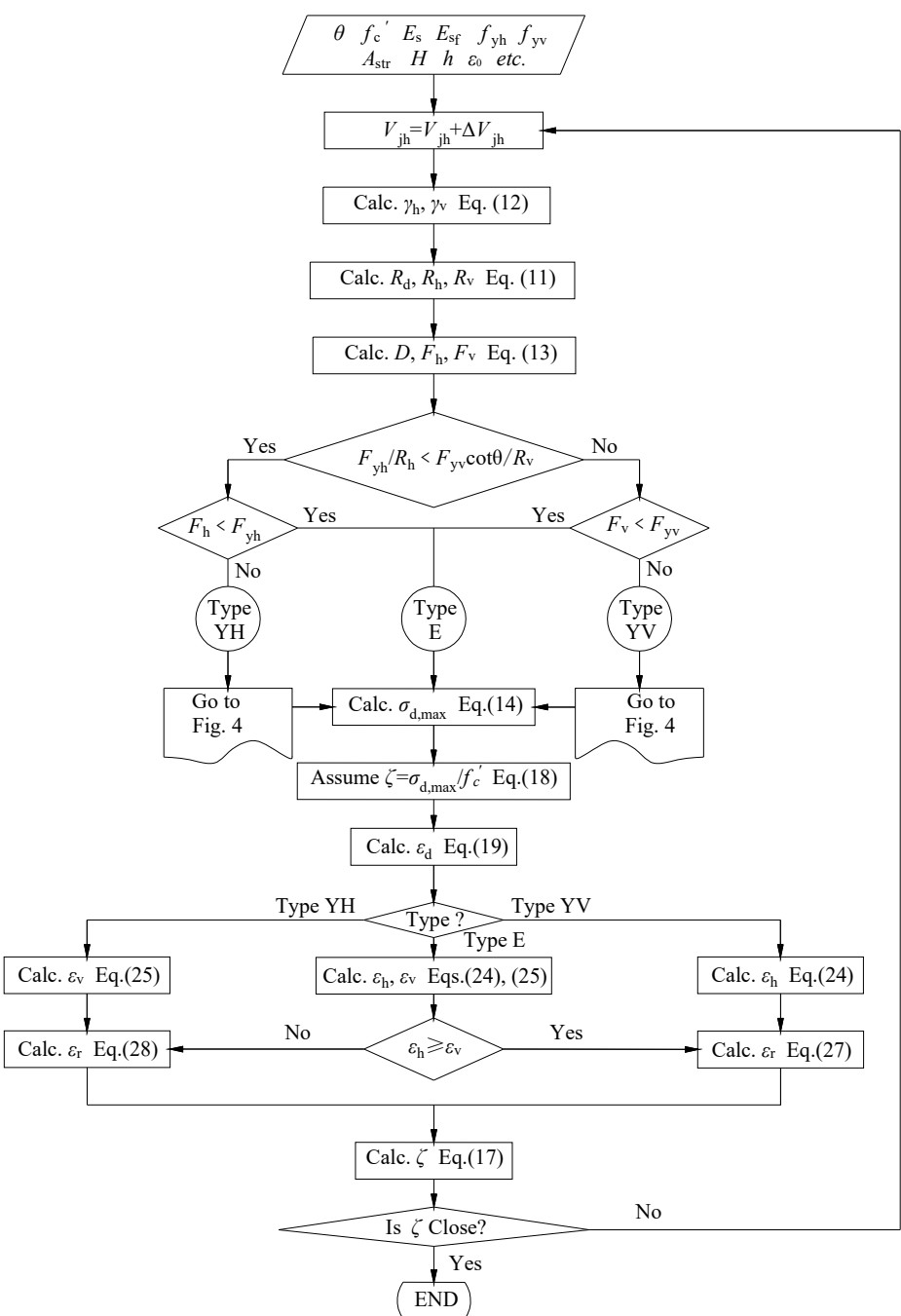

**Figure 3.** Flow chart showing the efficient algorithm.

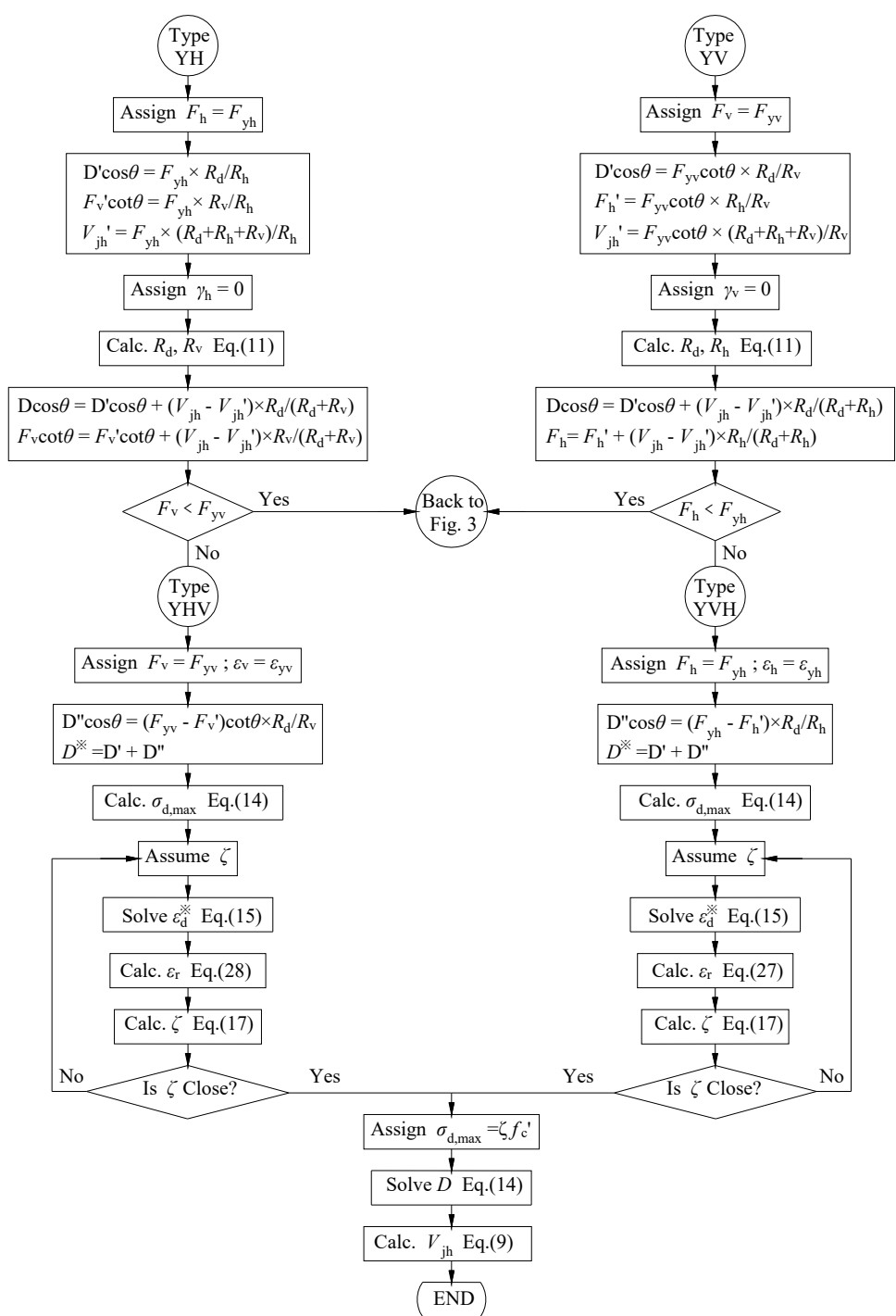

**Figure 4.** Algorithm for post-yielding cases.

**Table 1.** Calculation type.

| Type | Type E | Type YH | Type YV | Type YHV | Type YVH |
|------|--------|---------|---------|----------|----------|
| Condition | $F_h < F_{yh}$ and $F_v < F_{yv}$ | $F_h = F_{yh}$ and $F_v < F_{yv}$ | $F_h < F_{yh}$ and $F_v = F_{yv}$ | $F_h = F_{yh}$ and then $F_v = F_{yv}$ | $F_v = F_{yv}$ and then $F_h = F_{yh}$ |

### 4. Test Verification

In order to verify the SBC calculation model of SFRC SWs described above in this study, two SW specimens with 1/4 scale and a shear–span ratio of 1.0 were subjected to low-cycle repeated loading. This included one RHSC SW specimen and one SFHSC SW specimen. Table 2 shows the detailed properties of the SF. Table 3 shows the concrete mix proportion of the two test specimens. The test specimens considered herein have three major common features: (1) all walls showed a wall web shear-dominant failure mode; (2) they were one-story isolated walls; (3) all contained both horizontal and vertical reinforcement uniformly distributed basically throughout the wall web. The dimensions and reinforcement configurations of the two SW specimens designed and manufactured are shown in Figure 5. The reinforcement strength grade and the concrete strength grade of the SW specimen were HRB400 and C60, respectively. The SF was provided by Bekaert, Shanghai, China. Table 3 also presents the remaining design parameters of the SW specimens, including the volume ratio of SF, axial compression ratio, and concrete strength. After loading completely, the two SW specimens exhibited distinct shear failure characteristics, as illustrated in Figure 6, which depicts the failure patterns of the two SW specimens after the test. The test results indicated that the inclusion of SF significantly improved the crack formation and seismic behavior of the HRB400 level high-strength reinforced concrete SW specimen. As the volume ratio of SF increases, the cracks in the SW specimen become thinner and denser. The crack distribution area significantly increases, and the amount of concrete crushing and spalling of the wall web is reduced significantly in the final failure of the SW specimen. The test data of nine other SFRC SW specimens from the existing literature are also collected and utilized to validate the proposed method in this paper, as shown in Table 4. The SW specimens examined in this study encompass various test parameters, including the volume ratio of SF, reinforcement ratio, shear–span ratio, and concrete strength. All the tested SW specimens showed a shear failure mode.

The research in [44] gives the functional relationship between $f_c$ and $f_c'$. The last column of Table 4 provides the $V_{jh,t}/V_{jh,c}$ ratios, illustrating the computational accuracy of the proposed calculation method. The average strength ratio ($V_{jh,t}/V_{jh,c}$) is 0.958 with a coefficient of variation (COV) of 0.18 (refer to Table 2), and the results indicate that the calculated values are in excellent agreement with the experimental values for the 11 SFRC SW specimens mentioned above. The proposed calculation method is both scientific and accurate in analyzing and predicting the SBC of SFRC SWs. The SSTM can be applied to calculate the SBC of SFHSC and SFRC SWs.

**Table 2.** Properties of SF.

| SF Type | Equivalent Diameter (mm) | Length (mm) | Aspect Ratio | Tension Strength (MPa) | Elastic Modulus (GPa) |
|---------|--------------------------|-------------|--------------|------------------------|-----------------------|
| Hooked | 0.55 | 35 | 65 | 1345 | 200 |

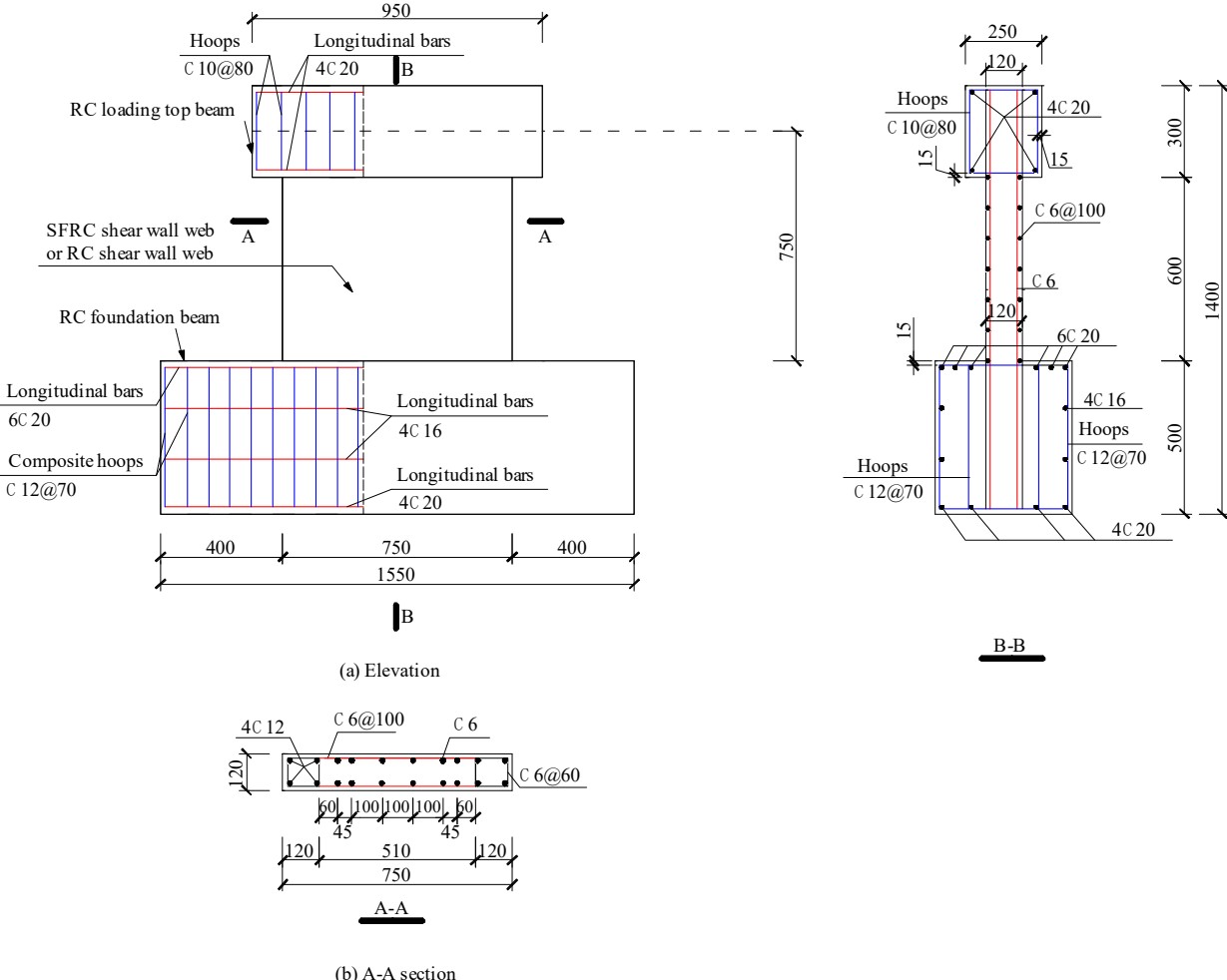

(a) Elevation

(b) A-A section

**Figure 5.** Dimension and reinforcement of shear wall specimens.

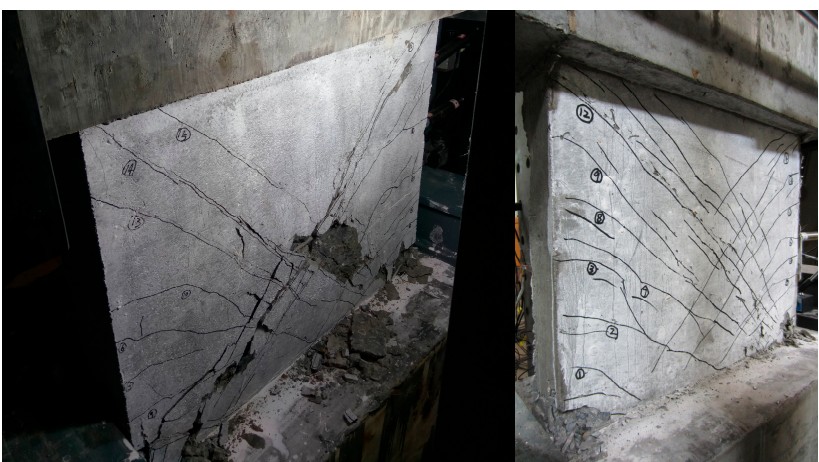

**Figure 6.** Failure patterns of shear wall specimens.

**Table 3.** The concrete mix proportion of the test specimens.

| Specimen | Strength Grade | Material Consumption/(kg·m$^{-3}$) | | | | | | $\rho_f$ |
|---|---|---|---|---|---|---|---|---|
| | | Cement | Water | Steel Fiber | Sand | Crushed Stone | Water-Reducing Agent | |
| RC-1.0-00-C60 | C60 | 529 | 164 | 0 | 646 | 1110 | 5.819 | 0 |
| RC-1.0-10(H)-CF60 | CF60 | 529 | 164 | 78 | 646 | 1110 | 5.819 | 1.0% |
| SW-05-40 [45] | CF40 | 454 | 168 | 39 | 676 | 1152 | —— | 0.5% |
| SW-10-40 [45] | CF40 | 476 | 176 | 78 | 719 | 1079 | —— | 1.0% |
| SW-15-40 [45] | CF40 | 503 | 186 | 117 | 740 | 1021 | —— | 1.5% |
| SW-20-40 [45] | CF40 | 524 | 194 | 156 | 779 | 953 | —— | 2.0% |
| SW-10-30 [45] | CF30 | 436 | 196 | 78 | 763 | 1054 | —— | 1.0% |
| FSW1 [46] | CF60-CF70 | 550 | 165 | 78 | 374 | 1326 | 5.5 | 1.0% |
| FSW2 [46] | CF60-CF70 | 550 | 165 | 117 | 369 | 1309 | 5.5 | 1.5% |
| FSW3 [46] | CF60-CF70 | 550 | 165 | 156 | 366 | 1299 | 5.5 | 2.0% |
| FSW4 [46] | CF60-CF70 | 550 | 165 | 78 | 374 | 1326 | 5.5 | 1.0% |

**Table 4.** Experimental verification.

| Specimen | $f_c$/MPa | $H$/mm | $b \times h$/mm | $n$ | Horizontal Reinforcement | | Vertical Reinforcement | | Steel Fiber | | $V_{jh,t}$/kN | $V_{jh,c}$/kN | $V_{jh,t}$/$V_{jh,c}$ |
|---|---|---|---|---|---|---|---|---|---|---|---|---|---|
| | | | | | Reinforcement | $f_{yh}$/MPa | Reinforcement | $f_{yv}$/MPa | $l_f$/$d_f$ | $\rho_f$/% | | | |
| RC-1.0-00-C60 | 55.4 | 750 | 120 × 750 | 0.2 | ϕ6@100 | 369.17 | 12ϕ6 | 369.17 | — | — | 546 | 552 | 0.989 |
| RC-1.0-10(H)-CF60 | 55.6 | 750 | 120 × 750 | 0.2 | ϕ6@100 | 369.17 | 12ϕ6 | 369.17 | 64 | 1.0 | 630 | 629 | 1.002 |
| SW-05-40 [45] | 21.2 | 900 | 200 × 900 | 0.1 | ϕ8@150 | 340 | 6ϕ14 | 373.5 | 57 | 0.5 | 730 | 608 | 1.201 |
| SW-10-40 [45] | 26.8 | 900 | 200 × 900 | 0.1 | ϕ8@150 | 340 | 6ϕ14 | 373.5 | 57 | 1.0 | 745 | 730 | 1.021 |
| SW-15-40 [45] | 25.1 | 900 | 200 × 900 | 0.1 | ϕ8@150 | 340 | 6ϕ14 | 373.5 | 57 | 1.5 | 770 | 748 | 1.029 |
| SW-20-40 [45] | 26.9 | 900 | 200 × 900 | 0.1 | ϕ8@150 | 340 | 6ϕ14 | 373.5 | 57 | 2.0 | 808 | 792 | 1.020 |
| SW-10-30 [45] | 17.8 | 900 | 200 × 900 | 0.1 | ϕ8@150 | 340 | 6ϕ14 | 373.5 | 57 | 1.0 | 730 | 586 | 1.246 |
| FSW1 [46] | 36.0 | 600 | 70 × 1000 | 0.09 | 6ϕ6.5 | 310 | 6ϕ6.5 | 310 | 64 | 1.0 | 335 | 431 | 0.777 |
| FSW2 [46] | 33.5 | 600 | 70 × 1000 | 0.09 | 6ϕ6.5 | 310 | 6ϕ6.5 | 310 | 64 | 1.5 | 330 | 436 | 0.757 |
| FSW3 [46] | 35.0 | 600 | 70 × 1000 | 0.09 | 6ϕ6.5 | 310 | 6ϕ6.5 | 310 | 64 | 2.0 | 340 | 465 | 0.731 |
| FSW4 [46] | 34.5 | 600 | 70 × 1000 | 0.09 | 6ϕ6.5 | 310 | 8ϕ6.5 | 310 | 64 | 1.0 | 330 | 430 | 0.767 |

## 5. Conclusions

In this paper, a scientific and accurate calculation method for determining the SBC of the SFRC SW is established. The proposed SSTM is based on the principles of struts and ties, and it satisfies the equilibrium, constitutive, and compatibility equations of cracked SFRC. Drawing from the test results of 11 low-rise SFRC SWs under low-cycle repeated loading in this study, along with data from the existing literature and a comparison with the established calculation method, the following conclusions can be made:

1. The two SFRC SW specimens in this paper exhibited obvious shear failure characteristics, and all SFRC SW specimens primarily showed a typical diagonal cracking pattern after the test. The inclusion of SF notably improved the crack formation and seismic behavior of the HRB400 level high-strength reinforced concrete SW specimen. With an increase in the volume ratio of SF, the cracks of the SW specimen became thinner and denser. The crack distribution area significantly increased, and the amount of concrete crushing and spalling of the wall web was significantly reduced in the final failure of the SW specimen.

2. The loading mechanism of SFRC SWs can be described by the SSTM. The SSTM of SFRC SWs, which consists of diagonal struts, horizontal, and vertical resistance members, has been established. This model distinguishes the contributions of SF, concrete, and distributed web reinforcement to the SBC of SFRC SWs.

3. The randomly distributed SF in the SW web can be equivalent to horizontal and vertical finely distributed steel bars in the SBC analysis of SFRC SWs, and the contributions of SF to the wall SBC are accurately predicted and identified.

4. After the experiments, the shear capacities of the two SFRC SW specimens in this paper are 546 kN and 630 kN, respectively, while the shear capacities calculated by the

SSTM model are 552 kN and 629 kN, respectively. The difference between the former and the experimental results is only 1.10%, and the latter differs from the experimental results by only 0.15%. This shows that the calculation results of the SSTM model proposed in this paper differ very little from the actual results, and the method can accurately calculate the shear bearing capacity of the specimen.

5.  In addition to the above experimental verification, we collected the shear capacity results of nine specimens from the literature for comparison with the calculated results. By using the proposed methodology, we obtained the calculated bearing capacity of these nine specimens and compared them with the known test results. The average strength ratio ($V_{jh,t}/V_{jh,c}$) is 0.95 with a COV of 0.18. The results show that the proposed calculation method is scientific and accurate for analyzing and predicting the SBC of SFRC SWs for diagonal compression failures.

The research on SFRC SWs based on SSTM still requires more simulation and experimental verification. Future studies can use more realistic structural models and more accurate experimental methods to further verify and improve the applicability and accuracy of the model. The ultimate goal is to apply the research results on the SBC of SFRC SWs under low-cycle repeated loading based on SSTM to practical engineering. Future research needs to strengthen communication and cooperation with engineering practices to transform research results into actual design specifications and construction guidelines.

**Author Contributions:** Conceptualization and original draft, P.Y.; methodology, J.Z., review and editing, B.W.; investigation and formal analysis, Y.W.; visualization, Q.Y.; validation and supervision, L.L. All authors have read and agreed to the published version of the manuscript.

**Funding:** This work was supported by the China National Natural Science Foundation Youth Fund Project (52108207) (52109168), Natural Science Foundation of Henan (212300410106), Young backbone teachers project of Henan University of Urban Construction (YCJQNGGJS202002), Academic and technical leaders project of Henan University of Urban Construction (YCJXSJSDTR202201), Innovation and Entrepreneurship Training Program for University Student (202211765031), The Training Plan for Young Backbone Teachers in Colleges and Universities of Henan Province, China (2023GGJS138), 2023 Key Scientific Research Project of Henan Province (23B560021), and 2023 Henan Province Key Research and Development and promotion Special project (science and technology) (232102320089).

**Data Availability Statement:** Data are contained within the article.

**Conflicts of Interest:** The authors declare no conflict of interest.

### Nomenclature

| | |
|---|---|
| SFRC | Steel-fiber-reinforced concrete |
| SW | shear wall |
| SSTM | softened strut-and-tie model |
| SF | steel fiber |
| SBC | shear bearing capacity |
| SFHSC | steel fiber reinforced high-strength concrete |
| RC | reinforced concrete |
| HSC | high-strength concrete |
| RHSC | reinforced high-strength concrete |

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
