# Peer review of "Shear Bearing Capacity of Steel-Fiber-Reinforced Concrete Shear Wall under Low-Cycle Repeated Loading Based on the Softened Strut-and-Tie Model"

_buildings, doi:10.3390/buildings14010012_

Round 1

Reviewer 1 Report

Comments and Suggestions for Authors

The paper needs revisions, particularly in the discussion sections.

Comments on the Quality of English Language

The manuscript should be revised for grammar and clarity; many of the used expressions are vague, and there are minor language errors

Reviewer 2 Report

Comments and Suggestions for Authors

This paper is discussing the use of fiber-reinforced concrete on shear wall capacity by applying the strut and tie model.

The methodology in this paper is not clear and the following notes were recorded.

1- The abstract is unclear and doe not describe the outcomes.

2- The introduction is very poor and further literature is required to describe the idea of the research.

3- Define the standard that was used for building the computation method.  Note that strut and tie are described in codes for deep beams.

4- The title needs to be changed to highlight that this paper is experimentally and numerically investigating the idea.

5- Conclusions are not clear and further work needs to be done there.  it is more like results rather than conclusions.

Comments on the Quality of English Language

The language is hard to understand while reading

Author Response

请参阅附件。

Reviewer 3 Report

Comments and Suggestions for Authors

This manuscript investigates the ‘Shear Bearing Capacity of SFRC Shear Wall based on SSTM. Please find the comments below.

·        Do not use abbreviations in the title.

·        Abstract: It directly started with your work. Before starting your aim, Mention the research background, novelty.

·        Introduction: in first line give the full form for SW and afterwards you can use the abbreviation SW.

·        State the research gap at the last paragraph of introduction.

Separate the section Discussion from the conclusion and strengthen the discussion section.

·        Add a separate section for nomenclature.

·        Mention the scope for future work.

·        Include more references by citing more works.

After carrying out a thorough review, it is recommended to do minor revisions

Round 2

Reviewer 2 Report

Comments and Suggestions for Authors

No further comments

Comments on the Quality of English Language

Language is fine